# The Effect of Alcohol Drinking on Metabolic Syndrome and Obesity in Koreans: Big Data Analysis

**DOI:** 10.3390/ijerph19094949

**Published:** 2022-04-19

**Authors:** Eun Jung Park, Hye Jung Shin, Sung Soo Kim, Ki Eun Kim, Sun Hyun Kim, Youl Ri Kim, Kyong Mee Chung, Kyung Do Han

**Affiliations:** 1Oksu Hana ENT Clinic, Seoul 04733, Korea; ejparkmd@gmail.com; 2Department of Pediatrics, National Medical Center, Seoul 04564, Korea; hyejungshin@empas.com; 3Department of Family Medicine, Chungnam National University Hospital, Chungnam National University College of Medicine, Daejeon 35015, Korea; 4Department of Pediatrics, CHA Gangnam Medical Center, CHA University School of Medicine, Seoul 06135, Korea; diavolezza@naver.com; 5Department of Family Medicine, International St. Mary’s Hospital, Catholic Kwandong University College of Medicine, Incheon 22711, Korea; sunhyun@yahoo.com; 6Department of Psychiatry, Seoul Paik Hospital, Inje University College of Medicine, Seoul 04551, Korea; youlri.kim@paik.ac.kr; 7Department of Psychology, Yonsei University, Seoul 03722, Korea; kmchung@yonsei.ac.kr; 8Department of Statistics and Actuarial Science, Soongsil University, Seoul 06978, Korea; hkd917@naver.com

**Keywords:** alcohol drinking, metabolic syndrome, obesity, big data

## Abstract

The purpose of this study was to assess the effect of alcohol consumption on metabolic syndrome (MetS) and obesity in Koreans by analysis of big data from the National Health Insurance Service health checkup database. A total of 26,991,429 subjects aged 20 years or older were included. Alcohol consumption was divided into five groups: nondrinkers, ≤7.0 g/d, 7.1–14.0 g/d, 14.1–28.0 g/d, ≥28.1 g/d. Logistic regression analyses were performed after adjusting for age, exercise, smoking, and income. The odds ratios (ORs) of MetS and obesity in men and women were lowest at ≤7.0 g/d, similar to that of the nondrinkers at 7.1–14.0 g/d, and increased with the alcohol consumption. At 7.1–14.0 g/d in older men, the ORs of metabolic syndrome and obesity were similar to those in the nondrinkers, but the OR of obesity was slightly increased in older women. This study suggests that the risk of MetS and obesity may be higher in Korean men, women, and the elderly who drink more than 14 g/d than the nondrinkers. In people with obesity or abdominal obesity, or those who need to manage their blood pressure, glucose, or triglyceride, drinking more than 7 g/d may increase the risk of MetS.

## 1. Introduction

Drinking is widespread all over the world. Alcohol consumption in the world, measured in liters of pure alcohol per person of 15 years of age or older, was 5.8 L in 2019. Koreans drink 8.45 L per capita per year, which is lower than the United States (9.97 L), the United Kingdom (11.45 L), and France (12.23 L), but above average [1]. Moderate alcohol consumption is not harmful to health, but excessive drinking increases the risk of cardiovascular disease, type 2 diabetes, dementia, and cancer [2]. In Korea, the prevalence of metabolic syndrome increased from 22.6% (male 25.1%, female 19.7%) in 2013 to 30.4% (male 32.2%, female 28.2%) in 2018 [3] and the prevalence of obesity increased from 32.6% in 2009 to 38.5% in 2018 [4]. Therefore, the effect of alcohol consumption on metabolic syndrome and obesity is important in managing these diseases.

Alcohol drinking is also well known as one of the risk factors for metabolic syndrome. In most studies of alcohol consumption and metabolic syndrome, heavy drinking is known to increase the risk of metabolic syndrome [5]. However, the alcohol consumption that increases the risk of metabolic syndrome differs among studies. The effect of drinking on metabolic syndrome in Koreans in the moderate drinking group is controversial because the results are different depending on the study [6,7].

Studies on the association between alcohol consumption and obesity are very diverse. In cross-sectional and prospective studies, it has been reported that alcohol consumption has a positive correlation with obesity or, on the contrary, has a negative correlation or no correlation. In addition, the results of the experimental studies on whether alcohol induces obesity are different from each other [8].

In Asians, including Koreans, alcohol metabolism is lower than in Caucasians due to mutations in aldehyde dehydrogenase (ALDH2), which is related to alcohol metabolism [9,10]. Therefore, even moderate drinking (28 g/d for men and 14 g/d for women and the elderly) as defined by the National Institute on Alcohol Abuse and Alcoholism (NIAAA) may increase the risk of metabolic syndrome in Koreans [11].

Among the large-scale studies on alcohol consumption and metabolic syndrome in Koreans, there are very few studies that classify alcohol consumption as less than one standard drink per day. Some of the studies that have been conducted are small-scale community-based studies or studies that focus on drinking patterns, so there are limitations in assessing the relationship between alcohol consumption and metabolic syndrome [12,13].

Thus, this study investigated the effect of alcohol drinking on metabolic syndrome and obesity using the National Health Insurance Service (NHIS) health checkup database after the moderate drinking group was subdivided.

## 2. Materials and Methods

### 2.1. The NHIS Database and NHIS Health Checkup Program

There are three programs of healthcare security system in South Korea: the National Health Insurance Program, the Medical Aid Program, and the Long-term Care Insurance Program. The National Health Insurance Program covers the whole population as a social insurance benefits scheme. The NHIS is a single insurer that manages the National Health Insurance Program, providing compulsory universal medical care to 96.3% of Koreans (2016) [14,15]. The National Health Insurance program provides a biannual health checkup program for all employees, except for those who must be screened annually.

The NHIS health checkup program includes self-reported questionnaires on demographic characteristics and health behavior, body mass index (BMI), waist circumference (WC), blood pressure (BP), fasting plasma glucose (FPG), total cholesterol, high density lipoprotein cholesterol (HDL-C), and triglyceride (TG), etc. Quality control procedures were in accordance with the Korean Association of Laboratory Quality Control and the hospitals that performed the health checkups were certified by the NHIS. Therefore, the NHIS database represents the entire Korean population and can be used as a population-based database [14].

### 2.2. Subjects

We included all adult men and women aged 20 or more who received NHIS health checkups from 2015 to 2016 (24 months) as subjects. A total of 26,991,429 subjects (14,516,804 men and 12,474,625 women) were recruited in this study.

### 2.3. Study Methods

The present study was designed as a cross-sectional study using big data of NHIS. Low-income groups (lower 20% or Medical Aid Program), large city dwellers (Seoul and metropolitan cities), past or present smoking, and regular exercise (moderate intensity/30 min or more/5 times or more per week, or severe intensity/20 min or more/3 times or more per week) were analyzed through self-reported questionnaires.

Alcohol consumption was assessed using a self-report questionnaire. This questionnaire included the amount of alcohol consumed at a time (1–2 drinks, 3–4 drinks, 5–6 drinks, 7–9 drinks, more than 10 drinks), frequency of drinking (non-drinking, 1/month, 2–4/month, 2–3/week, ≥4/week) and frequency of binge drinking (none, <1/month, 1/month, 1/week, daily). The amount of alcohol was computed as [amount of drinking (mL) × volume by alcohol (%) × specific gravity of ethanol at room temperature (0.79)]/100. Following the criteria of the NIAAA, an amount containing 14 g of alcohol was defined as one standard drink (drink afterwards) [11]. To calculate the daily drinking amount, the categories of the amount of alcohol consumed and frequency of drinking were converted into numbers. According to the NIAAA, moderate drinking is defined as up to 1 drink (14 g of alcohol) per day for women and up to 2 drinks (28 g of alcohol) per day for men [11]. The authors classified the amount of alcohol into 5 groups, including these standards (14 g and 24 g) and half drink (7 g) per day lower than these, as follows: nondrinkers, ≤7.0 g/d (half drink per day), 7.1–14.0 g/d (one drink per day), 14.1–28.0 g/d (two drinks per day), ≥28.1 g/d (more than two drinks per day).

According to the diagnostic criteria of the National Cholesterol Education Program Adult Treatment Panel III, metabolic syndrome was defined when at least 3 of the components of the metabolic syndrome were satisfied [16]. In detail, the diagnostic criteria of metabolic syndrome were as follows: WC ≥ 90 cm in men and ≥85 cm in women, TG ≥ 150 mg/dL, HDL-C ≤ 40 mg/dL in men and ≤50 mg/dL in women, systolic BP ≥ 130 mmHg or diastolic BP ≥ 85 mmHg, and FPG ≥ 100 mg/dL. If a drug was prescribed with a disease code for medical insurance claims for hypertension, diabetes, or dyslipidemia it was included as a component of the metabolic syndrome. The standards for WC of abdominal obesity and BMI for obesity were applied according to the guideline of the Korean Society for the Study of Obesity [17].

### 2.4. Statistical Methods

Continuous categorical variables were analyzed using the *t*-test and analysis of variance, whereas categorical variables were analyzed using the chi-squared test. A variable (TG) that did not satisfy the normal distribution was analyzed after log transformation. Logistic regression analysis was performed to evaluate the risk for metabolic syndrome after adjusting for age, exercise, smoking, and income. The statistical significance level was set to *p* = 0.0001. All statistical analyses were performed using SAS version 9.3 (SAS Institute, Cary, NC, USA).

## 3. Results

### 3.1. General Characteristics of Subjects

We analyzed 26,991,429 subjects (14,516,804 men and 12,474,625 women) aged 20 years or older. All variables were significantly different between men and women. The prevalence of obesity and abdominal obesity were significantly higher in men (40.33% and 24.44%, respectively) than in women (26.72% and 18.51%, respectively). The prevalence of metabolic syndrome was also significantly higher in men (14.91%) than in women (12.69%). The proportion of nondrinkers was significantly higher in women (71.62%) than in men (32.02%) (Table 1).

### 3.2. Prevalence of the Components of Metabolic Syndrome and Obesity According to Alcohol Consumption

Both men and women had the lowest prevalence of metabolic syndrome, obesity, and abdominal obesity in the group drinking 7.0 g/d or less; the prevalence of these increased according to the amount of alcohol consumed. In both men and women, the prevalence of BP, FPG, and TG increased as the amount of alcohol increased, but that of HDL-C decreased (Table 2).

### 3.3. Odds Ratios of Metabolic Syndrome and Obesity According to Alcohol Consumption

The odds ratio (OR) of metabolic syndrome significantly increased in proportion to alcohol consumption in both men and women. The OR of metabolic syndrome increased by at least 25% from 14.1 to 28.0 g/d for men and by at least 18% from 28 g/d or more for women compared to the nondrinkers. The OR of obesity increased by more than 10% from 7.1 to 14.0 g/d in men and more than 18% from 14.1–28.0 g/d in women. The OR of abdominal obesity (WC ≥ 90 cm in men and ≥ 85 cm in women) increased by more than 17% from 14.1–28.0 g/d in men and 7.1 to 14 g/d in women. BP component was at least 11% higher in all drinking groups than in nondrinkers in men and increased by at least 41% from 7.1 to 14.0 g/d in women. FPG component increased by at least 25% from 7.1 to 14.0 g/d in both men and women. TG component increased by at least 11% from 7.1 to 14.0 g/d in men and 14.1 to 28.0 g/d in women. The HDL-C component was lower in both men and women by more than 25% in the drinking group compared to the nondrinkers (Table 3).

### 3.4. Odds Ratios of Metabolic Syndrome and Obesity According to Age in Men

The OR of metabolic syndrome and obesity was lowest at 7.0 g/d or less in men aged 20 to 64 years and increased with alcohol consumption. However, in men over 65 years of age the OR of metabolic syndrome was 8% lower at 7.0 g/d or less, but it did not increase even if the amount of alcohol increased and there was no difference from the nondrinkers. However, the OR of obesity was highest at 15% at 14.1 to 28.0 g/d in men over 65 years of age. The ORs of abdominal obesity, BP, and FPG in men over 65 years of age increased with the amount of alcohol consumed (Table 4).

### 3.5. Odds Ratios of Metabolic Syndrome and Obesity According to Age in Women

The OR of metabolic syndrome and obesity in women aged 20 to 64 years increased with alcohol consumption similarly to men aged 20 to 64 years. In women over 65 years of age, the OR of metabolic syndrome was 9 to 15% lower in the drinking group than in the nondrinkers. In women over 65 years of age, the OR of obesity was 15% and 17% higher at 7.1 to 14.0 g/d and 14.1 to 28.0 g/d, respectively, than in the nondrinkers (Table 5).

## 4. Discussion

The purpose of this study was to assess the effect of alcohol consumption on metabolic syndrome and obesity in Koreans by analysis of big data from the National Health Insurance Service health checkup database. Due to the close relationship between alcohol drinking and metabolic syndrome and obesity in terms of these risks, it is clinically important to establish an appropriate amount of alcohol that is not harmful.

According to this study, the prevalence of metabolic syndrome among Koreans was confirmed to be 14.9% for men and 12.7% for women. This result is lower than the prevalence of metabolic syndrome (men 27.3%, women 20.2%) in 2015 (all examinees 5,986,920 men and 413,533 women) reported to the government by NHIS to suggest national policy directions for local health improvement [3]. Therefore, it is presumed that some of the patients receiving treatment for the underlying diseases of metabolic syndrome did not participate in the NHIS health checkup program due to various reasons such as duplicate laboratory tests or hospitalization, etc.

In a meta-analysis study by Sun et al., the risk of metabolic syndrome decreased (OR 0.86) at 5 g/d or less there was no significant difference from the nondrinkers at 5.1 to 35 g/d (subgroups: 5.1 to 10, 10.1 to 20, and 20.1 to 35 g/d; OR 1.07, 0.93, and 1.00, respectively), and increased when it exceeded 35 g/d. (OR 1.70) [1]. In a study by Baik et al., similar to the above, the risk of metabolic syndrome was not significant in moderate drinking (15.1 to 30 g/d) (RR 1.25 (0.75–2.090)), but significantly increased in excess of 30 g/d increase (RR 1.63 (1.02–2.62)) [7]. However, the other study showed that even light drinking (1–14.9 g/d) increased the risk of metabolic syndrome by 51% (OR 1.51) [6]. Most studies have confirmed that the risk of metabolic syndrome increases in heavy drinkers (>28 g/d); however, the risk of metabolic syndrome in moderate drinkers (≤28 g/d) is controversial.

The risk of metabolic syndrome was lowest at 7 g/d or less and increased in proportion to alcohol consumption in this study. The lowest risk of metabolic syndrome at 7 g/d or less was similar to that of Sun et al. (≤5 g/d) [5]. As alcohol consumption increases, the risk of metabolic syndrome increases even though the prevalence of HDL-C component (HDL-C ≤ 40 mg/dL in men and ≤50 mg/dL in women) also decreases (Table 2). This result could be explained by the fact that the protective effect of HDL-C is offset by the influence of other components of metabolic syndrome as the amount of drinking increases. An increase in HDL-C concentration during drinking is associated with an increase in the transport rate of the HDL apolipoprotein apoA-I and -II [18].

In our study, similar to the above two studies [5,7], heavy drinkers also increased the risk of metabolic syndrome by 42% and 18% in men and women, respectively. However, in moderate drinkers (14.1 to 28 g/d) the risk of metabolic syndrome increased by 25% and 7%, respectively, in men and women, unlike these two studies [5,7]. A meta-analysis study by Sun et al. [5] was conducted with a total of five studies (three studies were Caucasians, two studies were Koreans). The reason why the risk of metabolic syndrome did not increase in moderate drinkers in the above study [5] could be explained by the fact that Caucasians with better alcohol metabolism than Asians were included [19]. In a study of Baik et al. [7], the risk of metabolic syndrome was not significant in moderate drinkers (15.1 to 30 g/d) but showed a tendency to increase (RR 1.25). It could be assumed that more subjects in this study would have been statistically significant. In a study by Kim et al. [6], the risk of metabolic syndrome increased even in lighter drinkers (1–14.9 g/d). The present authors analyzed this drinking range by dividing it into 7 g/d or less and 7.1–14 g/d. The ORs of metabolic syndrome in the 7 g/d or less were 1.09 (1.08–1.09) in men and 0.97 (0.96–0.98) in women, similar to the results of Kim et al., [6]. However, at 7.1 to 14.0 g/d the risk of metabolic syndrome increased as described above. This result suggests that a subdivision lower than 14 g/d may be necessary when assessing the risk of metabolic syndrome in a race with a lower alcohol metabolism, such as Koreans.

Except for the HDL-C component of metabolic syndrome, the other four components increased with alcohol consumption. At 7.1–14.0 g/d, BP, FPG, and TG increased by 25%, 36%, and 17%, respectively, in men and BP, FPG, and WC increased by 41%, 27%, and 17%, respectively, in women. Therefore, in those who need control of BP, FPG, TG, and WC, even one drink per day (7.1 to 14.0 g/d) may increase the risk. Although the mechanism by which alcohol raises blood pressure is not yet clear, it can be explained by various mechanisms such as an imbalance of the central nervous system, impairment of the baroreceptors, increased sympathetic outflow, stimulation of the renin-angiotensin-aldosterone system, increased cortisol levels, increased intracellular calcium and vascular reactivity, stimulation of the endothelium to release vasoconstrictors (angiotensin II, endothelin-1, and norepinephrine), and loss of relaxation due to inhibition of endothelium-dependent nitric oxide production [20]. Drinking with food can raise plasma glucose, because alcohol is used as an energy source before food [21]. Alcohol-induced hypertriglyceridemia is due to increased very-low-density lipoprotein secretion, impaired lipolysis, and increased free fatty acid fluxes from adipose tissue to the liver [22].

In our study, the OR of metabolic syndrome in the elderly 65 years or older was similar to or slightly lower than that of the nondrinkers. As explained above, this reason is considered to be largely related to the omission of patients receiving treatment for the underlying disease of metabolic syndrome. In both elderly men and women, the risk of WC, FPG, and BP among the metabolic syndrome components is higher in the drinking groups than nondrinkers. Therefore, it may be considered that the risk of drinking still exists in the elderly.

In the present study, the reason that the minimum alcohol consumption, which increases the risk of metabolic syndrome, was lower than that of Caucasians could be explained by the lower alcohol metabolism in Asians. Hot flashes indicating alcohol sensitivity, which are about 3 to 29% in Caucasians, occur in 47 to 85% of Asians [19]. The mechanism of hot flashes is the higher accumulation of acetaldehyde in flushing subjects because they have an unusual less-active liver aldehyde dehydrogenase (ALDH) isozyme [22]. In the study analyzing the inactive form of ALDH as a phenotype, it was found that the ratio of the inactive form of ALDH2 in Koreans was 32% and that of Japanese people was 36% [23,24].

The results of studies on alcohol consumption and obesity or abdominal obesity are somewhat controversial. In a prospective study by Wannamethee et al., the heavy drinkers (>30 g/d) had a higher risk of weight gain than moderate drinkers or non-drinkers [25]. In a study by Sakurai et al. alcohol intake had a positive correlation with waist–hip ratio, but body mass index had no significant relationship [26]. Duncan et al. reported that moderate drinking reduced waist circumference [27]. However, in a nine-year prospective study by Dallongeville et al. there was no correlation between alcohol consumption and waist circumference [28]. The reason why the results of studies on the relationship between drinking alcohol and obesity are different is because the factors that cause obesity are very diverse. These factors include gender, amount and frequency of drinking, type of drinking (binge drinking), amount of physical activity, sleep patterns, depression, psychosocial problems, chronic diseases, drug use, eating habits that cannot control food, and genetic factors [8]. It is difficult to find a study that corrected all these confounding factors.

In this study, the risk of obesity and abdominal obesity in men and women, similar to the study by Duncan et al. [27], was slightly lower than the nondrinkers at 7 g/d or less, and then increased more than the nondrinkers according to the amount of alcohol consumed. The reason that the risk of obesity and abdominal obesity at 7 g/d or less was lower than that of the nondrinkers could be explained by the fact that it was related to a well-controlled lifestyle rather than the positive effect of alcohol when considering the results of the components of metabolic syndrome (Table 4 and Table 5). The risk of obesity or abdominal obesity increased by more than 10% at 7.1 to 14.0 g/d. At 14.1 to 28.0 g/d, the risk of both obesity and abdominal obesity increased by 18% or more. We suggest that the number of subjects in this study was able to significantly offset the effects of various confounding variables because the number of subjects in this study was representative of the entire population. Therefore, this result is expected to be helpful when educating patients on drinking alcohol who need to manage obesity and abdominal obesity.

This study has the following limitations. First, since this study is a cross-sectional study, the causality between alcohol consumption and metabolic syndrome and obesity cannot be determined. Some prospective studies have confirmed that heavy drinking may be a cause of metabolic syndrome [6,7]; therefore, a prospective study that subdivides moderate drinking in the future is necessary. Second, in the present study, alcohol consumption was assessed by self-reported questionnaire, so it may be underestimated compared to actual alcohol consumption; therefore, in future study an interview with a recent recall method is needed to more accurately evaluate the amount of alcohol consumed [29]. Third, there was no evaluation of nutrition among confounding factors. Considering that drinkers generally have higher total energy intake than nondrinkers [30], future studies need to accompany this evaluation. However, it is important that this study is the first and representative study of Koreans to analyze the relationship between drinking and metabolic syndrome with big data.

## 5. Conclusions

In agreement with all authors, the following criteria were developed to determine the amount of alcohol consumed that increases the risk of metabolic syndrome and obesity in men, women, and the elderly: (1) increased the OR of metabolic syndrome more than 10% compared to the nondrinkers (control group) or (2) increased three or more components of metabolic syndrome with at least one increased by more than 50%. According to these criteria, there is an increased risk of metabolic syndrome and obesity from alcohol drinking of 14.1 to 28 g/d in Korean men, women, and the elderly. Alcohol drinking of 7.1 to 14 g/d may increase the risk of metabolic syndrome and obesity in people requiring management of obesity, abdominal obesity, BP, FPG, or TG. Therefore, it may be necessary to determine the moderate amount of drinking in consideration of individual risk factors.

In conclusion, the present authors suggest that the risk of metabolic syndrome and obesity may be higher in Korean men, women, and elderly persons (65 years or older) who drink more than 14 g/d (one drink a day or seven drinks a week) than the nondrinkers. In people with obesity or abdominal obesity, or those who need to manage their BP, FPG, or TG, drinking more than 7 g/d (half drink a day or four drinks a week) may increase the risk of metabolic syndrome.

## Figures and Tables

**Table 1 ijerph-19-04949-t001:** Baseline characteristics of the subjects.

Variables	Male	Female	*p*-Value
Total number	14,516,804	12,474,625	
Age (year)	47.71 ± 13.7	50.19 ± 14.37	<0.0001
20 to 39	4,415,937 (30.42)	2,769,241 (22.20)	<0.0001
40 to 64	8,296,562 (57.15)	7,659,563 (61.40)	<0.0001
≥65	1,804,305 (12.43)	2,045,821 (16.40)	<0.0001
Low-income	2,105,800 (14.51)	2,889,503 (23.16)	<0.0001
Large city residents	6,286,419 (43.30)	5,685,870 (45.58)	<0.0001
Past or present smokers	10,023,308 (69.05)	669,206 (5.36)	<0.0001
Regular exercise	3,367,994 (23.20)	2,252,416 (18.06)	<0.0001
Height (cm)	170.53 ± 6.49	157.08 ± 6.23	<0.0001
Weight (kg)	71.26 ± 10.99	57.27 ± 8.88	<0.0001
Waist circumference (cm)	84.29 ± 8.11	76.48 ± 9.14	<0.0001
Abdominal obesity	3,547,448 (24.44)	2,309,114 (18.51)	<0.0001
BMI (kg/m^2^)	24.45 ± 3.15	23.22 ± 3.46	<0.0001
Obesity (BMI ≥ 25)	5,853,936 (40.33)	3,332,743 (26.72)	<0.0001
Underweight (BMI < 18.5)	287,186 (1.98)	695,849 (5.58)	<0.0001
SBP (mm Hg)	124.23 ± 13.38	119.01 ± 14.98	<0.0001
DBP (mm Hg)	77.68 ± 9.41	73.71 ± 9.68	<0.0001
FPG (mg/dL)	101.42 ± 24.84	96.48 ± 20.49	<0.0001
T-C (mg/dL)	193.64 ± 36.75	195.26 ± 37.07	<0.0001
TG (mg/dL) *	123.55 (123.51–123.59)	92.24 (92.21–92.27)	<0.0001
HDL-C (mg/dL)	52.09 ± 13.33	59.81 ± 14.68	<0.0001
LDL-C (mg/dL)	113.43 ± 35.97	114.15 ± 34.98	<0.0001
Metabolic syndrome	2,164,785 (14.91)	1,582,426 (12.69)	<0.0001
Drinking			<0.0001
Nondrinkers	4,648,007 (32.02)	8,934,014 (71.62)	
≤7.0 g/d	3,194,505 (22.01)	2,446,858 (19.61)	
7.1 to 14.0 g/d	2,306,251 (15.89)	563,425 (4.52)	
14.1 to 28.0 g/d	2,321,398 (15.99)	347,776 (2.79)	
≥28.1 g/d	2,046,643 (14.1)	182,552 (1.46)	
Binge drinking †			<0.0001
No	13,665,462 (94.14)	12,272,085 (98.38)	
Yes	851,342 (5.86)	202,540 (1.62)	

*p* values were obtained by *t*-test for continuous variables and by chi-squared test for categorical variables. Data are expressed as number (%) or mean ± standard error. *—geometric mean (95% confidence interval), †—binge, 5 or more drinks in a 2 h time frame in men, 4 or more in women, low-income group— lower 20% or Medical Aid Program, large city residents—residents of Seoul or metropolitan cities, regular exercise—moderate intensity/30 min or more/5 times or more per week or severe intensity/20 min or more/3 times or more per week, abdominal obesity—waist circumference ≥ 90 cm in men and ≥ 85 cm in women, BMI—body mass index, SBP—systolic blood pressure, DBP—diastolic blood pressure, FPG—fasting plasma glucose, T-C—total cholesterol, TG—triglyceride, HDL-C—high-density lipoprotein cholesterol, LDL—low-density lipoprotein cholesterol.

**Table 2 ijerph-19-04949-t002:** Prevalence of the components of metabolic syndrome and obesity according to alcohol consumption.

Variables	Nondrinkers	≤7.0 g Per Day	7.1 to 14.0 g Per Day	14.1 to 28.0 g Per Day	≥28.1 g Per Day	*p*-Value
Men						
WC	23.28 (23.24–23.32)	21.78 (21.73–21.82)	24.26 (24.21–24.32)	26.28 (26.23–26.34)	29.32 (29.26–29.38)	<0.0001
BP	43.8 (43.78–43.87)	46.18 (46.13–46.23)	51.33 (51.27–51.4)	55.68 (55.62–55.74)	59.38 (59.31–59.45)	<0.0001
FPG	36.65 (36.6–36.69)	37.99 (37.93–38.04)	41.49 (41.43–41.55)	45.34 (45.28–45.4)	48.79 (48.72–48.85)	<0.0001
HDL-C	31.11 (31.08–31.15)	25.42 (25.38–25.47)	24.42 (24.36–24.47)	22.94 (22.89–23.00)	21.81 (21.75–21.86)	<0.0001
TG	40.45 (40.41–40.5)	39.70 (39.65–39.76)	44.05 (43.99–44.12)	47.87 (47.81–47.93)	51.17 (51.11–51.24)	<0.0001
MetS	14.08 (14.05–14.11)	13.05 (13.01–13.08)	14.85 (14.81–14.9)	16.35 (16.31–16.4)	18.14 (18.1–18.19)	<0.0001
Obesity	38.41 (38.36–38.45)	37.35 (37.3–37.41)	40.76 (40.69–40.82)	43.24 (43.18–43.31)	45.53 (45.46–45.6)	<0.0001
Women						
WC	18.46 (18.44–18.49)	17.28 (17.24–17.33)	20.64 (20.54–20.74)	22.19 (22.06–22.32)	23.73 (23.56–23.91)	<0.0001
BP	35.38 (35.35–35.41)	35.90 (35.85–35.96)	41.48 (41.37–41.59)	44.99 (44.84–45.13)	47.75 (47.55–47.95)	<0.0001
FPG	28.44 (28.41–28.47)	28.84 (28.78–28.89)	32.85 (32.74–32.97)	35.28 (35.13–35.42)	37.11 (36.90–37.31)	<0.0001
HDL-C	38.39 (38.36–38.42)	30.95 (30.89–31.01)	29.49 (29.37–29.6)	28.27 (28.12–28.42)	27.90 (27.69–28.10)	<0.0001
TG	30.18 (30.15–30.21)	26.35 (26.29–26.40)	29.57 (29.46–29.69)	32.09 (31.94–32.23)	35.02 (34.82–35.21)	<0.0001
MetS	12.87 (12.85–12.89)	11.49 (11.45–11.53)	13.33 (13.25–13.42)	14.02 (13.91–14.12)	15.01 (14.86–15.15)	<0.0001
Obesity	26.88 (26.85–26.91)	24.99 (24.93–25.04)	28.52 (28.41–28.64)	29.84 (29.7–29.99)	30.54 (30.34–30.74)	<0.0001

*p* values were obtained by analysis of variance. Data are expressed as % (95% confidence interval). WC—waist circumference, BP—blood pressure, FPG—fasting plasma glucose, HDL-C—high-density lipoprotein cholesterol, TG—triglyceride, MetS—metabolic syndrome, obesity—BMI ≥ 25 kg/m^2^.

**Table 3 ijerph-19-04949-t003:** Odds ratios of metabolic syndrome and obesity according to alcohol consumption.

Amount	WC	BP	FPG	HDL-C	TG	MetS	Obesity
Men							
Nondrinkers	1	1	1	1	1	1	1
≤7.0 g/d	0.92 (0.91–0.92)	1.11 (1.1–1.12)	1.06 (1.06–1.07)	0.75 (0.75–0.75)	0.97 (0.97–0.97)	0.91 (0.90–0.91)	0.96 (0.96–0.96)
7.1–14.0 g/d	1.06 (1.05–1.06)	1.39 (1.38–1.39)	1.25 (1.24–1.25)	0.71 (0.71–0.72)	1.17 (1.16–1.17)	1.09 (1.08–1.09)	1.10 (1.10–1.11)
14.1–28.0 g/d	1.18 (1.17–1.18)	1.67 (1.67–1.68)	1.47 (1.47–1.48)	0.66 (0.66–0.66)	1.36 (1.36,1.37)	1.25 (1.24–1.25)	1.22 (1.22–1.23)
≥28 g/d	1.37 (1.36–1.37)	1.96 (1.95–1.97)	1.70 (1.70–1.71)	0.62 (0.62–0.62)	1.56 (1.55–1.56)	1.42 (1.42–1.43)	1.34 (1.34–1.35)
*p*-value	<0.0001	<0.0001	<0.0001	<0.0001	<0.0001	<0.0001	<0.0001
Women							
Nondrinkers	1	1	1	1	1	1	1
≤7.0 g/d	0.89 (0.89–0.89)	1.03 (1.03–1.04)	1.02 (1.01–1.02)	0.68 (0.67–0.68)	0.77 (0.77–0.77)	0.76 (0.76–0.77)	0.89 (0.89–0.89)
7.1–14.0 g/d	1.17 (1.16–1.18)	1.41 (1.40–1.42)	1.27 (1.26–1.28)	0.60 (0.59–0.60)	0.94 (0.93–0.94)	0.97 (0.96–0.98)	1.09 (1.08–1.10)
14.1–28.0 g/d	1.32 (1.30–1.33)	1.73 (1.71–1.74)	1.45 (1.44–1.46)	0.55 (0.54–0.55)	1.11 (1.10–1.12)	1.07 (1.06–1.09)	1.18 (1.17–1.19)
≥28 g/d	1.46 (1.45–1.48)	2.00 (1.98–2.02)	1.59 (1.58–1.61)	0.52 (0.51–0.52)	1.31 (1.29–1.33)	1.18 (1.15–1.20)	1.22 (1.20–1.23)
*p*-value	<0.0001	<0.0001	<0.0001	<0.0001	<0.0001	<0.0001	<0.0001

Odds ratios (95% confidence interval) were obtained by logistic regression analysis after adjusting for age, income, smoking status, exercise. Amount—gram of alcohol per day (g/d), WC—waist circumference, BP—blood pressure, FPG—fasting plasma glucose, HDL-C—high-density lipoprotein cholesterol, MetS—metabolic syndrome, obesity—body mass index ≥ 25 kg/m^2^.

**Table 4 ijerph-19-04949-t004:** Odds ratios of metabolic syndrome and obesity according to age in men.

Amount	WC	FPG	BP	HDL-C	TG	MetS	Obesity
Age 20–39							
Nondrinkers	1	1	1	1	1	1	1
≤7.0 g/d	0.92 (0.91–0.93)	1.00 (0.99–1.00)	1.04 (1.04–1.05)	0.70 (0.70–0.71)	0.94 (0.94–0.95)	0.82 (0.81–0.83)	0.96 (0.96–0.97)
7.1–14.0 g/d	1.05 (1.05–1.06)	1.13 (1.13–1.14)	1.27 (1.27–1.28)	0.65 (0.64–0.65)	1.15 (1.15–1.16)	1.01 (0.99–1.02)	1.12 (1.11–1.13)
14.1–28.0 g/d	1.18 (1.17–1.19)	1.32 (1.31–1.33)	1.54 (1.53–1.55)	0.55 (0.54–0.55)	1.38 (1.37–1.39)	1.15 (1.14–1.17)	1.26 (1.25–1.26)
≥28 g/d	1.40 (1.39–1.41)	1.53 (1.52–1.54)	1.86 (1.85–1.88)	0.49 (0.48–0.49)	1.69 (1.68–1.70)	1.41 (1.40–1.43)	1.44 (1.43–1.45)
*p*-value	<0.0001	<0.0001	<0.0001	<0.0001	<0.0001	<0.0001	<0.0001
Age 40–64							
Nondrinkers	1	1	1	1	1	1	1
≤7.0 g/d	0.90 (0.90–0.91)	1.06 (1.06–1.06)	1.15 (1.14–1.15)	0.74 (0.74–0.74)	0.96 (0.96–0.97)	0.90 (0.90–0.91)	0.94 (0.93–0.94)
7.1–14.0 g/d	1.05 (1.04–1.05)	1.25 (1.25–1.26)	1.46 (1.45–1.46)	0.71 (0.71–0.71)	1.15 (1.15–1.16)	1.09 (1.08–1.10)	1.07 (1.07–1.08)
14.1–28.0 g/d	1.17 (1.16–1.17)	1.48 (1.47–1.48)	1.76 (1.75–1.77)	0.66 (0.66–0.67)	1.33 (1.32–1.33)	1.25 (1.24–1.25)	1.17 (1.16–1.17)
≥28 g/d	1.37 (1.36–1.38)	1.72 (1.71–1.73)	2.05 (2.04–2.06)	0.64 (0.63–0.64)	1.50 (1.49–1.50)	1.43 (1.43–1.44)	1.28 (1.28–1.29)
*p*-value	<0.0001	<0.0001	<0.0001	<0.0001	<0.0001	<0.0001	<0.0001
Age ≥ 65							
Nondrinkers	1	1	1	1	1	1	1
≤7.0 g/d	0.97 (0.96–0.97)	1.03 (1.02–1.04)	1.14 (1.13–1.15)	0.78 (0.77–0.79)	0.95 (0.95–0.96)	0.92 (0.92–0.93)	0.99 (0.98–1.00)
7.1–14.0 g/d	1.10 (1.08–1.11)	1.17 (1.15–1.18)	1.41 (1.40–1.43)	0.71 (0.70–0.72)	1.04 (1.03–1.05)	1.00 (0.99–1.02)	1.08 (1.07–1.10)
14.1–28.0 g/d	1.20 (1.19–1.22)	1.28 (1.27–1.29)	1.58 (1.56–1.60)	0.66 (0.65–0.67)	1.08 (1.07–1.10)	1.04 (1.03–1.05)	1.15 (1.14–1.16)
≥28 g/d	1.22 (1.21–1.24)	1.42 (1.39–1.42)	1.66 (1.64–1.68)	0.55 (0.55–0.56)	1.10 (1.09–1.11)	1.01 (0.99–1.02)	1.08 (1.07–1.09)
*p*-value	<0.0001	<0.0001	<0.0001	<0.0001	<0.0001	<0.0001	<0.0001

Odds ratios (95% confidence interval) were obtained by logistic regression analysis after adjusting for age, income, smoking status, exercise. Amount—gram of alcohol per day (g/d), WC—waist circumference, BP—blood pressure, FPG—fasting plasma glucose, HDL-C—high-density lipoprotein cholesterol, MetS—metabolic syndrome, obesity—body mass index ≥ 25 kg/m^2^.

**Table 5 ijerph-19-04949-t005:** Odds ratios of metabolic syndrome and obesity according to age in women.

Amount	WC	FPG	BP	HDL-C	TG	MetS	Obesity
Age 20–39							
Nondrinkers	1	1	1	1	1	1	1
≤7.0 g/d	0.74 (0.74–0.75)	1.04 (1.04–1.05)	1.01 (1.00–1.02)	0.71 (0.70–0.71)	0.67 (0.66–0.67)	0.75 (0.74–0.77)	0.83 (0.82–0.83)
7.1–14.0 g/d	0.98 (0.97–1.00)	1.29 (1.27–1.31)	1.39 (1.37–1.41)	0.61 (0.60–0.62)	0.92 (0.90–0.93)	1.03 (0.99–1.07)	1.05 (1.03–1.13)
14.1–28.0 g/d	1.08 (1.06–1.10)	1.46 (1.43–1.48)	1.68 (1.65–1.71)	0.50 (0.49–0.51)	1.17 (1.15–1.20)	1.03 (0.99–1.08)	1.11 (1.10–1.25)
≥28 g/d	1.25 (1.22–1.27)	1.64 (1.60–1.67)	2.01 (1.97–2.05)	0.45 (0.44–0.46)	1.49 (1.46–1.52)	1.20 (1.14–1.27)	1.23 (1.20–1.25)
*p*-value	<0.0001	<0.0001	<0.0001	<0.0001	<0.0001	<0.0001	<0.0001
Age 40–64							
Nondrinkers	1	1	1	1	1	1	1
≤7.0 g/d	0.94 (0.94–0.95)	1.04 (1.04–1.04)	1.06 (1.06–1.07)	0.68 (0.68–0.68)	0.83 (0.82–0.83)	0.82 (0.81–0.83)	0.93 (0.93–0.94)
7.1–14.0 g/d	1.25 (1.24–1.26)	1.33 (1.32–1.34)	1.43 (1.42–1.44)	0.60 (0.60–0.60)	1.00 (0.99–1.01)	1.06 (1.05–1.08)	1.15 (1.14–1.16)
14.1–28.0 g/d	1.43 (1.41–1.44)	1.54 (1.52–1.56)	1.76 (1.74–1.77)	0.57 (0.57–0.58)	1.27 (1.16–1.19)	1.21 (1.19–1.23)	1.25 (1.23–1.26)
≥28 g/d	1.53 (1.51–1.56)	1.71 (1.68–1.73)	1.95 (1.92–1.98)	0.56 (0.55–0.57)	1.36 (1.34–1.38)	1.34 (1.31–1.37)	1.24 (1.22–1.26)
*p*-value	<0.0001	<0.0001	<0.0001	<0.0001	<0.0001	<0.0001	<0.0001
Age ≥ 65							
Nondrinkers	1	1	1	1	1	1	1
≤7.0 g/d	1.06 (1.04–07)	0.94 (0.932–0.95)	1.00 (0.99–1.02)	0.71 (0.70–0.72)	0.84 (0.83–0.85)	0.85 (0.84–0.86)	1.05 (1.03–1.06)
7.1–14.0 g/d	1.22 (1.18–1.26)	1.12 (1.08–1.15)	1.20 (1.16–1.25)	0.60 (0.58–0.62)	0.82 (0.79–0.85)	0.91 (0.88–0.94)	1.15 (1.11–1.19)
14.1–28.0 g/d	1.27 (1.22–1.26)	1.17 (1.12–1.22)	1.30 (1.23–1.36)	0.53 (0.51–0.55)	0.82 (0.79–0.86)	0.91 (0.87–0.95)	1.17 (1.12–1.22)
≥28 g/d	1.25 (1.18–1.34)	1.25 (1.17–1.33)	1.31 (1.22–1.41)	0.46 (0.43–0.49)	0.79 (0.75–0.85)	0.85 (0.80–0.91)	1.02 (0.96–1.09)
*p*-value	<0.0001	<0.0001	<0.0001	<0.0001	<0.0001	<0.0001	<0.0001

Odds ratios (95% confidence interval) were obtained by logistic regression analysis after adjusting for age, income, smoking status, exercise. Amount—gram of alcohol per day (g/d), WC—waist circumference, BP—blood pressure, FPG—fasting plasma glucose, HDL-C—high-density lipoprotein cholesterol, MetS—metabolic syndrome, obesity—body mass index ≥ 25 kg/m^2^.

## Data Availability

The NHIS database can only access permitted data only to those who have applied for the use of data for research in advance and have been approved.

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
