# Peer review of "The Effect of Alcohol Drinking on Metabolic Syndrome and Obesity in Koreans: Big Data Analysis"

_ijerph, 2022, doi:10.3390/ijerph19094949_

Round 1

Reviewer 1 Report

This cross-sectional study aimed to assess the effect of alcohol drinking on metabolic syndrome and obesity using big data of NHIS. Alcohol intake is a much underestimated problem, not least because of the ambiguous information disseminated by nutrition experts on the possible health-promoting effect of moderate alcohol consumption (e.g. red wine). This study is conducted in a formally good manner and shows interesting information in a clear and simple way. 

There are some issues to be improved before publication:

  1. there are some plagiarised parts, please correct them (see attached file)
  2. Are there differences between wine (red in particular) and other alcoholic drinks in the risk of metabolic syndrome and obesity?
  3. The authors should consider confounding factors. For example, it is possible that alcohol consumption is only a proxy for other factors (e.g. excessive caloric intake, reduced fibre consumption, etc.).
  4. The introduction is sterile. A few paragraphs on the harmful effects of alcohol consumption and its effects on glycidic metabolism should be included.

Author Response

Thanks for the detailed comment. The authors have made the corrections as mentioned.

This cross-sectional study aimed to assess the effect of alcohol drinking on metabolic syndrome and obesity using big data of NHIS. Alcohol intake is a much underestimated problem, not least because of the ambiguous information disseminated by nutrition experts on the possible health-promoting effect of moderate alcohol consumption (e.g. red wine). This study is conducted in a formally good manner and shows interesting information in a clear and simple way. 

There are some issues to be improved before publication:

  1. there are some plagiarised parts, please correct them (see attached file)

Answer: We made the modifications as requested, especially primary source 1 (line 145-146, 154-156, 159-162, 529-540), 3 (line 78-81, 91-92), 4 (389-393), 5(line 139-143, 148-151), and 8 (line 143-144).

  1. Are there differences between wine (red in particular) and other alcoholic drinks in the risk of metabolic syndrome and obesity?

Answer: The type of alcohol was not separated, only the amount of alcohol was calculated. Since the permit period has expired and data cannot be accessed again, analysis according to the type of alcohol was not possible.

  1. The authors should consider confounding factors. For example, it is possible that alcohol consumption is only a proxy for other factors (e.g. excessive caloric intake, reduced fibre consumption, etc.).

Answer: Data on nutrition could not be analyzed because these data were not available. We added this to the limitations of this study (line 471-473).

Third, there was no evaluation of nutrition among confounding factors. Considering that drinkers generally have higher total energy intake than nondrinkers [31], future studies need to accompany this evaluation.

  1. The introduction is sterile. A few paragraphs on the harmful effects of alcohol consumption and its effects on glycidic metabolism should be included.

Answer: Comparison of alcohol consumption by country, harmful effects of alcohol, prevalence of metabolic syndrome and obesity, and the relationship between alcohol consumption and obesity were added to the introduction (line 35-44, 53-57).

Line 35-44: Drinking is widespread all over the world. Alcohol consumption in the world, measured in liters of pure alcohol per person of 15 years of age or older, was 5.8 liters in 2019. Koreans drink 8.45 liters per capita per year, which is lower than the United States (9.97 liters), United Kingdom (11.45 liters) and France (12.23 liters), but above average [1]. Moderate alcohol consumption is not harmful to health, but excessive drinking increases the risk of cardiovascular disease, type 2 diabetes, dementia, and cancer [2]. In Korea, the prevalence of metabolic syndrome increased from 22.6% (male 25.1%, female 19.7%) in 2013 to 30.4% (male 32.2%, female 28.2%) in 2018 [3], and the prevalence of obesity increased from 32.6% in 2009 to 38.5% in 2018 did [4]. Therefore, the effect of alcohol consumption on metabolic syndrome and obesity is important in managing these diseases.

Line 53-57: Studies on the association between alcohol consumption and obesity are very diverse. In cross-sectional and prospective studies, it has been reported that alcohol consumption has a positive correlation with obesity or, on the contrary, has a negative correlation or no correlation. In addition, the results of the experimental studies on whether alcohol induces obesity are different from each other [8].

Reviewer 2 Report

The paper is well-written, precise, and structured according to standards. The dataset of enormous size makes this study unique and particularly important for practitioners. Used statistical methods are appropriate and performed according to guidelines. The researchers clearly listed methods and limitations. The language in the paper is unambiguous. Overall, the article can be presented in its current form.

Minor comments:

It is unclear whether there were empty fields left in questionnaires when the responders skipped some of the answers. Percent of non-responders for each question should be listed.

Page 3. Lines 98-100. It might be helpful to describe abbreviations at the first mention as the reader first encounters this line before Table 1.

Page 11. Extra character in reference 12.

Author Response

Reviewer 2

Thanks for the detailed comment. The authors have made the corrections as mentioned.

The paper is well-written, precise, and structured according to standards. The dataset of enormous size makes this study unique and particularly important for practitioners. Used statistical methods are appropriate and performed according to guidelines. The researchers clearly listed methods and limitations. The language in the paper is unambiguous. Overall, the article can be presented in its current form.

Minor comments:

It is unclear whether there were empty fields left in questionnaires when the responders skipped some of the answers. Percent of non-responders for each question should be listed.

Answer: Since it is mandatory to submit a questionnaire when receiving an examination, most of them are recorded. Because the current data access period has ended, the percentage of questionnaire omissions cannot be checked again.

Page 3. Lines 98-100. It might be helpful to describe abbreviations at the first mention as the reader first encounters this line before Table 1.

Answer: Since we described the full name and abbreviation in 2.1 (line 83-86), we only describe the abbreviation in 2.3.

Line 83-86: The NHIS health checkup program includes self-reported questionnaires on demographic characteristics and health behavior, body mass index (BMI), waist circumference (WC), blood pressure (BP), fasting plasma glucose (FPG), total cholesterol, high density lipoprotein cholesterol (HDL-C) and triglyceride (TG), etc.

Page 11. Extra character in reference 12.

Answer: We made the modifications as requested as follows (line 580-582).

Line 580-582: Seo, M.H.; Lee, W.Y.; Kim, S.S.; Kang, J.H.; Kang, J.H.; Kim, K.K.; Kim, B.Y.; Kim, Y.H.; Kim, W.J.; Kim, E.M.; et al. 2018 Korean
Society for the Study of Obesity Guideline for the Management of Obesity in Korea. J. Obes. Metab. Syndr. 2019, 28, 40–45.https://doi.org/10.7570/jomes.2019.28.1.40

Reviewer 3 Report

This review is interesting and important for understanding the effect of alcohol drinking on metabolic syndrome and obesity. However, the manuscript must be revised or supplemented for clarity.

Introduction Section

(Comment 1) I recommend authors to supplement statistical informations of metabolic syndrome, alcohol drinking and obesity in "Introduction Section"

(Comment 2) I recommend authors to supplement association between alcohol drinking and obesity in "Introuduction Section".

(Comment 3) Why the author express '26,991,429 subjects' as big data? I recommend authors to supplement this point in "Introuduction Section". (line 52)

Materials and Methods

(Comment 4) Sample selection process is not fully described. I recommend authors to supplement this point in "2.2 subjects". (line 73)

(Comment 5) The authors combined alcohol drinking quantity and frequency, and converted it into alcohol intake. Is there any reference of evidence to explaing this part? It is suspected that this part is divided according to the results after data analysis. (line 83-94)

Discussion Section

(Comment 6) For the prevalence differences, I recommend authors to add brief information about the size, subject, and method of each study (Korea National Statistical Office and NHIS health checkup program). Based on this, I recommend authors to explain the main reasons for the difference in prevalence.

(Comment 7) I recommend authors to suggest detail suggestions for future researchers.

Author Response

Thanks for the detailed comment. The authors have made the corrections as mentioned.

This review is interesting and important for understanding the effect of alcohol drinking on metabolic syndrome and obesity. However, the manuscript must be revised or supplemented for clarity.

Introduction Section

(Comment 1) I recommend authors to supplement statistical informations of metabolic syndrome, alcohol drinking and obesity in "Introduction Section"

Answer: Comparison of alcohol consumption by country, harmful effects of alcohol, prevalence of metabolic syndrome and obesity were added to the introduction (line 35-44).

(Comment 2) I recommend authors to supplement association between alcohol drinking and obesity in "Introuduction Section".

Answer: the association between alcohol drinking and obesity were added to the introduction (line 53-57).

(Comment 3) Why the author express '26,991,429 subjects' as big data? I recommend authors to supplement this point in "Introuduction Section". (line 52)

Answer: It has been modified as follows (line 69-71).

Thus, this study investigated the effect of alcohol drinking on metabolic syndrome and obesity using the National Health Insurance Service (NHIS) health checkup database after the moderate drinking group was subdivided.

Materials and Methods

(Comment 4) Sample selection process is not fully described. I recommend authors to supplement this point in "2.2 subjects". (line 73)

Answer: It has been modified as follows (line 92-93).

We included all adult men and women aged 20 or more who received NHIS health checkup from 2015 to 2016 (24 months) as subjects.

(Comment 5) The authors combined alcohol drinking quantity and frequency, and converted it into alcohol intake. Is there any reference of evidence to explaing this part? It is suspected that this part is divided according to the results after data analysis. (line 83-94)

Answer: We tried to analyze the effects of alcohol by including the NIAAA-mentioned moderate drinking 14g (women) and 24g (men), and also adding 7g, which is less than 14g. We've made up for what we didn't explicitly state as follows (line 127-134).

Line 127-134: To calculate the daily drinking amount, the categories of the amount of alcohol and frequency of drinking were converted into number. According to the NIAAA, moderate drinking is defined as up to 1 drink (14 g of alcohol) per day for women and up to 2 drinks (28 g of alcohol) per day for men [11]. The authors classified amount of alcohol into 5 groups, including these standards (14g and 24g) and half drink (7g) per day lower than these, as follows: nondrinkers, ≤ 7.0 g/d (half drink per day), 7.1-14.0 g/d (one drink per day), 14.1-28.0 g/d (two drinks per day), ≥ 28.1 g/d (more than two drinks per day).

Discussion Section

(Comment 6) For the prevalence differences, I recommend authors to add brief information about the size, subject, and method of each study (Korea National Statistical Office and NHIS health checkup program). Based on this, I recommend authors to explain the main reasons for the difference in prevalence.

Answer: It has been modified as follows (line 301-304).

Line 301-304: This result is lower than the prevalence of metabolic syndrome (men 27.3%, women 20.2%) in 2015 (all examinees 5,986,920 men and 4,13,533 women) reported to the government by NHIS to suggest national policy directions for local health improvement [3].  

(Comment 7) I recommend authors to suggest detail suggestions for future researchers.

Answer: For the first and second limitations, the future research direction has already been described, and after adding the third limitation, future research directions are presented. It has been modified as follows (line 463-473).

Line 463-473: First, since this study is a cross-sectional study, the causality between alcohol consumption and metabolic syndrome and obesity cannot be determined. Some prospective studies have confirmed that heavy drinking may be a cause of metabolic syndrome [6,7]. Therefore, a prospective study that subdivides moderate drinking in the future is necessary. Second, in the present study, alcohol consumption was assessed by self-reported questionnaire, so it may be underestimated compared to actual alcohol consumption. Therefore, in future study, an interview with a recent recall method is needed to more accurately evaluate the amount of alcohol consumed [30]. Third, there was no evaluation of nutrition among confounding factors. Considering that drinkers generally have higher total energy intake than nondrinkers [31], future studies need to accompany this evaluation.  

Round 2

Reviewer 1 Report

The authors responded to my comments. The paper is now publishable

Reviewer 3 Report

After reviewing the author responses, the author successfully addressed most of the comments and suggestions.